# Anti-SARS-CoV-2 serology based on ancestral RBD antigens does not correlate with the presence of neutralizing antibodies against Omicron variants

Léa Dépéry,[1] Isabelle Bally,[1] Axelle Amen,[1,2] Benjamin Némoz,[1] Marlyse Buisson,[1,3] Laurence Grossi,[1] Aurélie Truffot,[1,3] Raphaële Germi,[1,3] Delphine Guilligay,[1] Mélanie Veloso,[4] Antoine Vilotitch,[4] Olivier Epaulard,[1,5] Patrice Morand,[1,3] Winfried Weissenhorn,[1] Pascal Poignard,[1,3] Julien Lupo[1,3]

**ABSTRACT**  Neutralizing antibody titers and binding antibody levels are considered correlates of protection against severe SARS-CoV-2 infection. The clinical utility of serology should be reevaluated in light of the emergence of escape variants, as commercial antibody-binding assays have not been adapted to the virus' antigenic evolution. We compared anti-SARS-CoV-2 antibody titers in four quantitative serological tests based on variable ancestral spike antigens (three in-house ELISAs and the prototype VIDAS SARS-CoV-2 IgG QUANT assay) and neutralization assays against the pseudotyped Wuhan, BA.2, BA.4/5, BQ.1.1, and XBB.1.1 viruses in a cohort of 100 patients infected in 2020 or during the Omicron waves. Binding antibody levels correlated well with neutralizing antibody titers for Wuhan, BA.2, and BA.4/5, but the association decreased for BQ.1.1 and XBB.1 (for the VIDAS assay, Spearman's correlation was 0.82 [95% CI 0.74–0.88] and 0.61 [0.46–0.72] for BA.2 and XBB.1, respectively). In 15% of patients with no neutralizing antibodies against XBB.1, the VIDAS assay still yielded binding antibody levels ranging from 74 to 7,652 binding antibody units/mL. Using an adjusted threshold based on receiver operating characteristic (ROC) curve analysis, the specificity of neutralizing antibody detection increased from 0.15 (95% CI 0.02–0.45) and 0.17 (0.04–0.41) to 0.92 (0.64–1.00) and 0.83 (0.59–0.96) against BQ.1.1 and XBB.1, respectively. Serological tests based on receptor-binding domain antigens from the ancestral virus fail to predict neutralizing activity against the latest circulating Omicron variants. Adapting serological tests may improve their clinical utility in immunocompromised patients.

**IMPORTANCE**  Anti-SARS-CoV-2 serology was developed in 2020 in response to the COVID-19 pandemic to diagnose SARS-CoV-2 infection and monitor an individual's immunity following natural infection or vaccination. Given the relationship between neutralizing antibody titers and protection against severe infection, many studies have evaluated the correlation between serology tests and neutralization assays in the pre-Omicron era. An important potential clinical use of serology, which explores binding antibodies, is estimating an individual's level of protection against new infection, particularly in immunosuppressed individuals and those at risk of severe COVID. However, in the Omicron era, as new viruses evade the immunity induced by previous infections and vaccination, the correlation between binding antibody levels determined by serological assays developed from ancestral antigens and neutralizing antibody titers against new viruses should be re-examined in order to determine whether these assays should be optimized by adapting antigens to the circulating SARS-CoV-2 strains.

**KEYWORDS**  SARS-CoV-2, serology, neutralizing antibodies, Omicron, COVID

Address correspondence to Julien Lupo, jlupo@chu-grenoble.fr.

The virology laboratory (CHU Grenoble Alpes) received grants from bioMérieux to carry out the study.

See the funding table on p. 12.

S ince the emergence of the SARS-CoV-2 pandemic in 2020, the virus has evolved under selection pressure from the immune system. New viruses have been selected with numerous mutations in the spike glycoprotein, particularly in the receptor-binding domain (RBD) region, a major target of the neutralizing antibody response. Compared with the ancestral Wuhan strain, the Omicron variants carry more than 100 mutations, giving them a selective advantage in evading the immune response induced by previous infections, vaccination, or monoclonal antibody therapies (1). Consequently, the antibody response of a patient infected by Omicron viruses targets different epitopes from that of a patient infected in 2020 (2).

Anti-SARS-CoV-2 serology was developed in 2020 to diagnose SARS-CoV-2 infection and monitor an individual's immunity following natural infection or vaccination. Most commercial tests are based on the spike or RBD antigens from the ancestral Wuhan strain. An important potential clinical use of serology, which explores binding antibodies, is estimating an individual's level of protection against new infection, particularly in immunosuppressed individuals and those at risk of severe COVID (3–5). Given the relationship between neutralizing antibody titers and protection against severe infection, many studies have evaluated the correlation between serology tests and neutralization assays (6–9). A good to moderate correlation was reported between binding antibody levels and neutralizing antibody titers against Wuhan, Alpha, Delta, or Omicron BA.1/BA.2 viruses (10–17), although few studies have evaluated this correlation against the recent Omicron variants (escape viruses appeared after 2022 in Europe). Although neutralizing antibody titers are a clear correlate of protection, identifying a protective threshold applicable to a serological test is challenging. This challenge is further complicated by the continual emergence of new escape variants, whereas serological tests have not been adapted to the antigenic evolution of the virus.

In our study, we sought to determine whether serological tests based on spike or RBD antigens from the ancestral SARS-CoV-2 strain are still suitable for predicting the neutralizing activity against Omicron variants of sera from patients infected in the Omicron era. To address this question, we investigated a cohort of 100 patients infected in 2020 or during the successive Omicron waves and compared the results of anti-SARS-CoV-2 antibody titers between four different quantitative serological tests based on ancestral spike antigens, including an assay developed for commercial use, and neutralization assays against the pseudotyped Wuhan, BA.2, BA.4/5, BQ.1.1, and XBB.1.1 variants.

## MATERIALS AND METHODS

### Patients

We retrospectively selected a cohort of 100 patients infected at different times during the SARS-CoV-2 pandemic. The cohort included 20 patients infected with the ancestral Wuhan strain during the first wave of the pandemic (March–April 2020) (12) and 80 patients infected with Omicron variants between April 2022 and August 2023 ($n$ = 20 BA.2, $n$ = 20 BA.4/5, $n$ = 20 BQ.1.1, and $n$ = 20 XBB-infected patients). Sera from patients infected in 2020 were collected 6 months after primary infection, while sera from those infected with Omicron variants were collected between 3 weeks and 2 months after PCR diagnosis and sequencing results. The main patient characteristics (age, sex, sequencing results, vaccination status, etc.) are shown in Table S1. Sera were treated at 56°C for 30 minutes (decomplementation), aliquoted, and then stored at −20°C or +4°C depending on the time of re-use for analysis. Subjects were all informed and did not oppose; written consent was not required for the retrospective study in accordance with the national legislation and the institutional requirements.

### Cell culture

The pseudoviruses were produced using HEK293T cells. For titration and serum neutralization, we used HeLa-ACE2 cells (Scripps Research, La Jolla, USA), which were

genetically modified to overexpress the ACE2 receptor. Both cell types were maintained in culture in DMEM medium (Gibco, 31966-021) supplemented with 10% SVF (Dutscher, 500105), 1% penicillin-streptomycin, and 1% L-glutamine (Life Technologies, 10378016) and kept in the incubator at 37°C with 5% $CO_2$.

## Plasmids

The production of pseudoviruses required gag/pol, luciferase plasmids (kindly provided by P. Charneau, Pasteur, France), and plasmid coding for the spike protein of interest (Wuhan plasmid, D. Nemazee, Scripps Research, La Jolla, USA; BA.2 and BA.4/5 plasmids, InvivoGene [plv-spike-V12 and plv-spike-V13, respectively]; BQ.1.1 and XBB.1 plasmids, Addgene, references 194493 and 194494, respectively).

## Anti-SARS-CoV-2 antibody control

Different positive controls were used for the neutralization assays to validate our experiments and assess our inter-assay reproducibility: CC12.1 monoclonal antibody (Scripps Research, La Jolla, USA) at 6 µg/mL or Evusheld (tixagevimab + cilgavimab, AstraZeneca) at 2 µg/mL for the neutralization of Wuhan virus, cilgavimab or Evusheld at 2 µg/mL for the neutralization of BA.2 and BA.4/5 viruses, and two patient sera (diluted 1:10) neutralizing BQ.1.1 and XBB.1 viruses.

## Pseudo-typed SARS-CoV-2 virion production

Neutralization assays were performed using lentiviral pseudotypes harboring the SARS-CoV-2 spike and encoding luciferase. Briefly, gag/pol, luciferase plasmids, and the desired SARS-CoV-2 full-length spike plasmid were co-transfected on adherent HEK293T cells using Helix-in (OZ Biosciences, HX11000). Supernatants containing the produced pseudoviruses were harvested 72 h after transfection, centrifuged, filtered through 0.45 µm, concentrated 10 times on Amicon Ultra (MWCO 100 KDa), aliquoted, and stored at −80°C. Before use, the supernatants were titered using HeLa ACE-2 cells to determine the appropriate dilution of pseudovirus necessary to obtain about 200,000 relative light units per well in a 96-well plate.

## Neutralization assays

Serial threefold dilutions starting from 1/10 dilution (serum) or control antibody were incubated with pseudoviruses at 37°C for 1 h and then transferred onto HeLa-ACE2 cells in 96-well plates at 8,000 cells/well (Corning). Plates were incubated at 37°C for 48 h, and HeLa-ACE2 cells were further lysed using 1× luciferase lysis buffer (Oz Biosciences) at room temperature for 1 h. Luciferase activity was measured by adding luciferase substrate (Oz Biosciences) according to the manufacturer's instructions. Luciferase intensity was then read using a TECAN luminometer. The results from this assay were expressed as the serum dilution required to reduce infection by 50% (neutralization titer). Titers >30 were considered positive (18).

## Protein production and purification

Full-length spike (FL-S), spike without RBD (ΔRBD-S), and RBD proteins were used as antigens derived from the ancestral Wuhan strain in ELISA experiments. Protein production and purification were carried out as previously described (19). ΔRBD-S protein was kindly provided by Prof. Winfried Weissenhorn (Institut de Biologie Structurale, Grenoble, France).

## In-house quantitative ELISA

The antigens, either ancestral FL-S, ΔRBD-S, or RBD, were diluted to 1 µg/mL in PBS, and 50 µL/well was coated overnight at 4°C in 96-well plates (Maxisorp NUNC Immunoplate, 442404). Plates were washed using a ThermoScientific Microplate washer (5165040).

After three washes with 100 μL of PBS-Tween 0.05%, plates were blocked with PBS-BSA 3% for 1 h. Serum dilutions were added to each well for 1 h at room temperature. Plates were then washed three times with PBS-Tween 0.05%. A goat anti-human Fc IgG secondary antibody linked to alkaline phosphatase (Jackson Immuno, 109056098), diluted to 1/10,000 in PBS-BSA 1%, was added for 1 h. Following three washes, 50 μL of p-NitroPhenyl Phosphate (Interchim, UP 664791) was added. After 1 h at 37°C, absorbance was read at 405 nm with a TECAN Spark microplate reader. Antibody titers were expressed as ED50 (effective dilution 50 values) and determined as the serum dilution at which IgG binding was reduced by 50%. ED50 was calculated from crude data (O.D) after normalization using GraphPad Prism (version 9) "log(inhibitor) vs normalized response" function.

## VIDAS SARS-CoV-2 IgG QUANT

In-house ELISA and neutralization assays were compared with a non-commercialized assay from bioMerieux (prototype VIDAS SARS-CoV-2 IgG QUANT). The VIDAS SARS-CoV-2 IgG QUANT is an automated assay using the enzyme-linked fluorescent assay for the quantitative measurement of IgG antibodies to SARS-CoV-2. This test uses RBD as an antigen from the ancestral strain, as its previous version was based on qualitative results (20). It is calibrated against the WHO international standard, and the level of IgG antibodies quantified in serum is expressed in binding antibody units (BAU/mL), as defined in the context of the first wave of infection (aid in COVID-19 diagnosis). The detection limit of the test was 11 BAU/mL, and the cutoff value was set at 25 BAU/mL. As the linearity range was from 25 to 1,500 BAU/mL, the serum (1/10 and more) had to be diluted when the signal was above 1,500 BAU/mL. All experiments were performed on a VIDAS instrument according to the manufacturer's recommendations (bioMerieux).

## Statistical analyses

Median, interquartile range (IQR), 95% confidence intervals (CI 95%), correlation, and linear regression analyses were determined using GraphPad Prism 9 (GraphPad Software, Inc.) and Stata 15 (StataCorp, College Station, TX, USA). Serum neutralization curves were modeled using Python, with a four-parameter curve of the type $[(A − D)/\{1 + [(x/C) × B]\}] + D$. Antibody titers were compared using the non-parametric Wilcoxon signed-rank test or the Wilcoxon rank-sum test when appropriate. Differences in frequencies were compared using Fisher's exact test. Correlation between ELISA IgG titers and neutralizing antibody titers was performed using Spearman's correlation. Agreement between neutralization assays and serological tests was estimated using Cohen's kappa coefficient. Cutoff points for IgG levels or titers were optimized using ROC curve analysis and the Youden index (or Liu index for comparison between neutralizing antibody titers against BA.4.5 and VIDAS IgG levels). The threshold for the significance of the results was set at 0.05. Bonferroni correction was also used to account for multiple comparisons.

## RESULTS

### Omicron variants evade neutralizing antibody response from previous infection

Patient sera were tested against pseudotyped Wuhan, BA.2, BA.4/5, BQ.1.1, and XBB.1 viruses. Considering the global population, which included patients infected in 2020 and after the BA.2, BA4/5, BQ.1, and XBB breakthroughs, we showed that the level of the neutralizing antibody response decreased when measured against the successive Omicron waves. The neutralization titers relative to BA.2 virus were ~0.5 and ~2- to sixfold lower for BA.4/5, BQ.1.1, and XBB.1 viruses, respectively ($P < 0.001$ for all three viruses) (Fig. 1A). Accordingly, almost all patients (98%) had detectable neutralizing antibodies against the ancestral virus compared to the XBB.1 virus (82%) ($P < 0.05$). Focusing our analysis on the patient subgroups (Fig. S1), the median neutralizing

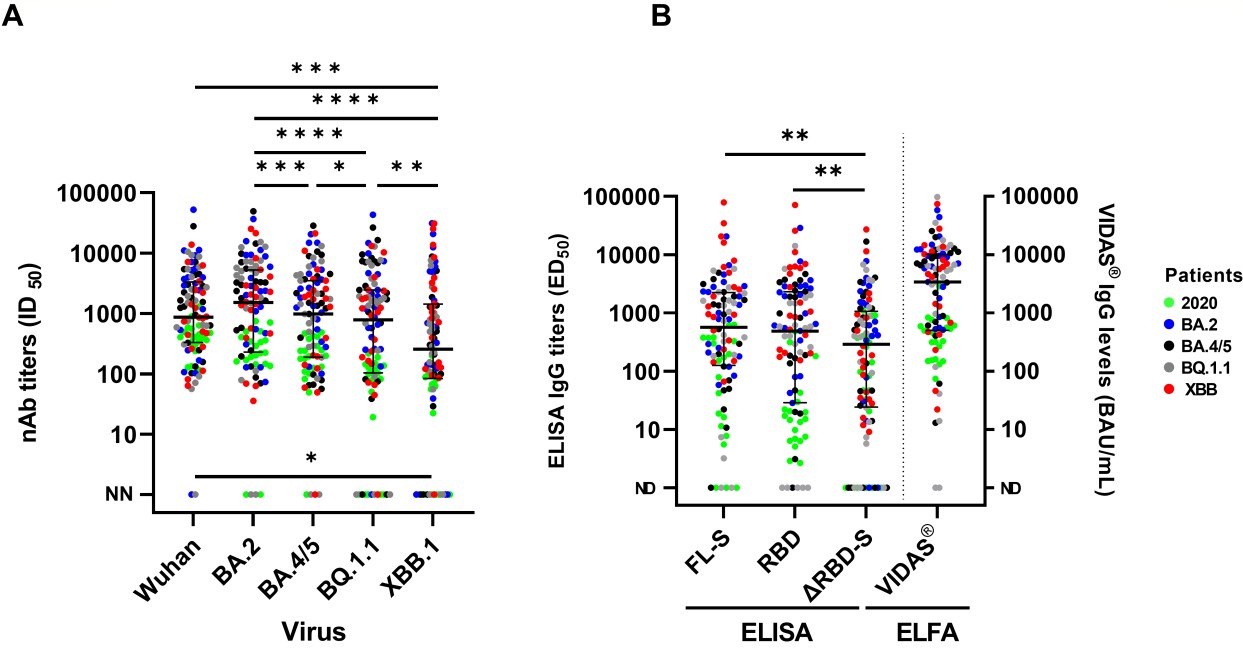

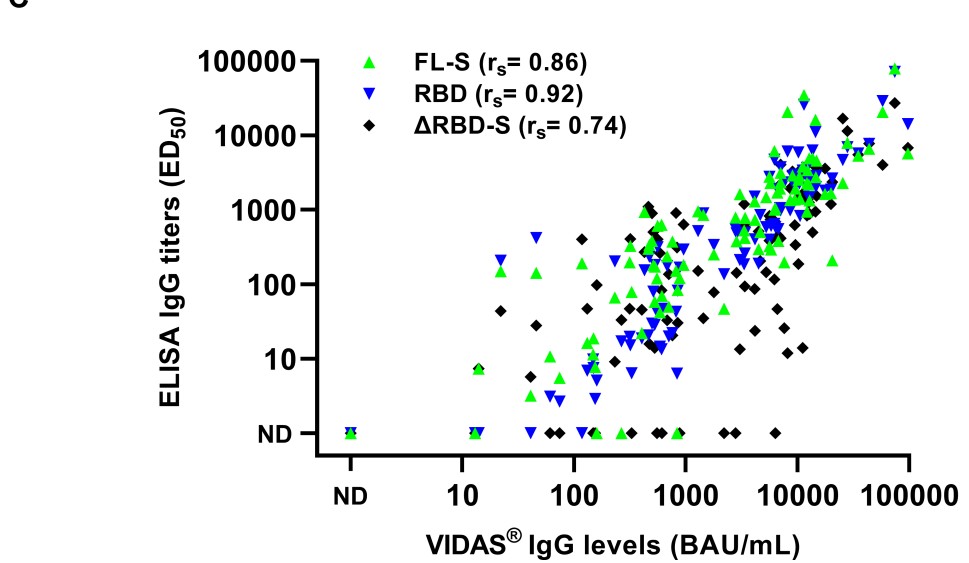

**FIG 1** Results of the neutralization assays and serological tests in the global cohort. (A) Neutralization assays against the Wuhan, BA.2, BA.4/5, BQ1.1, and XBB.1 viruses. (B) Results of the in-house ELISAs and prototype VIDAS anti-SARS-CoV-2 IgG QUANT assay. (C) Agreement between the in-house ELISA and VIDAS assays. Horizontal lines represent the median antibody titer values with the interquartile range. nAb, neutralizing antibody; NN, non-neutralizing antibody; ND, not detectable; ELFA, enzyme-linked fluorescent assay; $r_s$, Spearman's correlation. Green circles: 2020 patients; blue circles: BA.2 patients; black circles: BA.4/5 patients; gray circles: BQ1.1 patients; red circles: XBB patients. ****$P < 0.0001$; ***$P < 0.001$; **$P < 0.01$; and *$P < 0.05$.

antibody titer for patients infected in 2020 ranged from 403.9 (IQR, 319.6–530.9) against the ancestral virus to 68.6 (IQR 0–100.7) against the XBB.1 virus ($P < 0.001$). Neutralizing antibody titers against the Wuhan virus were higher in patients with BA.2 (1,606, IQR 480.2–3,698.2), BA.4/5 (3,449.7, IQR 389.9–6,376.1), BQ1.1 (1,570.9, IQR 450.4–4,729.8),

and XBB (844.3, IQR 361.4–2,334.5) infections than in those infected in 2020 (403.9, IQR 319.6–530.9; $P = 0.004$, 0.006, 0.01, and 0.033, respectively).

## Binding antibodies against ancestral SARS-CoV-2 spike antigens

Anti-SARS-CoV-2 IgG antibodies were tested using three different in-house ELISAs and the prototype VIDAS SARS-CoV-2 IgG QUANT assay, all based on ancestral spike antigens. Median IgG titers were lower for the in-house ELISA based on the non-RBD spike (290.9, IQR 24.7–1,223.0) than for the ELISA based on RBD (487.1, IQR 29.0–2,298; $P < 0.01$) and FL-S antigens (569.1 IQR 131.9–2,222; $P < 0.01$) (Fig. 1B). The frequency of undetectable antibodies for the ΔRBD-S ELISA was 15% and significantly higher than for the VIDAS assay (2%, $P < 0.05$), RBD ELISA (6%, $P < 0.05$), and FL-S ELISA (6%, $P < 0.05$). For the VIDAS assay and the in-house ELISAs, IgG levels were significantly lower in patients infected in 2020 than those infected after the Omicron breakthrough ($P < 0.01$). The VIDAS assay showed excellent correlation with FL-S and RBD ELISA (Spearman's correlation $r_s = 0.86$ [95% CI 0.80–0.91] and 0.92 [0.88–0.94], respectively). The correlation between FL-S and RBD ELISA was also high ($r_s = 0.91$ [95% CI 0.87–0.94]), whereas it was significantly lower between ΔRBD-S ELISA and ELISA based on FL-S, RBD, and VIDAS assay ($r_s = 0.68$ [95% CI 0.56–0.77], 0.66 [0.53–0.76], and 0.74 [0.63–0.82], respectively) (Fig. 1C).

## Correlation between neutralizing antibody titers and binding antibody levels

Spearman's correlations between VIDAS IgG levels and neutralizing antibody titers against Wuhan, BA.2, BA.4/5, BQ.1.1, and XBB.1 viruses were 0.76 (95% CI 0.66–0.83), 0.82 (0.75–0.88), 0.77 (0.68–0.84), 0.63 (0.50–0.74), and 0.61 (0.47–0.72), respectively (Fig. 2A; Table S2). The VIDAS assay thus showed a lower correlation with neutralization against BQ.1.1 and XBB.1 viruses (values not significantly different from the threshold of 0.6, considered to be a moderate correlation) (21). The in-house ELISA based on FL-S and RBD antigens showed similar results to the VIDAS assay. ΔRBD-S serology showed a weaker correlation than other tests with neutralization assays, but unlike the other tests, the correlation results did not decrease with BQ1 and XBB.1 variants (Fig. 2A; Table S2). We also assessed the agreement between VIDAS and neutralization assays using scatter plot diagrams (Fig. 2B through D). The agreement between VIDAS IgG levels and neutralizing antibody titers against BQ.1.1 and XBB.1 viruses was low (Cohen's kappa $\kappa = 0.16$ [95% CI 0–0.43] and 0.20 [0–0.43], respectively), with a greater dispersion of points around the linear regression line compared with Wuhan, BA.2, and BA4/5 viruses (linear regression coefficient $R^2 = 0.29$ and 0.30 vs 0.60, 0.67, and 0.57, respectively). In 11 and 15 patients with undetectable neutralizing antibody titers (<30) against BQ.1.1 and XBB.1 viruses, respectively, the VIDAS assay gave discrepant results with positive antibody levels ranging from 74 to 7,652 BAU/mL (Fig. 2E).

## Performance of serological tests to predict neutralizing activity

As we observed a lower correlation and agreement between serology and neutralization against BQ.1.1 and XBB.1 viruses, we investigated the possibility of using thresholds to improve the ability of binding antibody assays to predict the neutralizing activity of sera. We first focused our analysis on the prototype commercial assay. Using the supplier's cutoff of 25 BAU/mL, the specificity of the VIDAS assay for the presence of neutralizing antibodies to BQ.1.1 and XBB.1 viruses was 0.15 (95% CI 0.02–0.45) and 0.17 (0.04–0.41), respectively (Fig. 3A). When the cutoff was set using the Youden index at 4,324.5 and 865.5 BAU/mL, the specificity of neutralizing antibody detection against BQ.1.1 and XBB-1 virus significantly increased to 0.92 (95% CI 0.64–1) and 0.83 (0.59–0.96), respectively (Table S3). However, using this threshold, sensitivity decreased from 0.97 (95% CI 0.90–0.99) and 0.98 (0.91–1) to 0.53 (0.42–0.64) and 0.71 (0.60–0.80) for BQ.1.1 and XBB.1, respectively. The agreement between VIDAS serology and neutralization was even better before the Omicron era (Wuhan virus, $\kappa = 0.80$ [95% CI 0.40–1]; BQ.1.1, $\kappa = 0.20$ [0.07–0.32], $P < 0.01$; XBB.1, $\kappa = 0.37$ [0.20–0.55], $P < 0.05$) (Fig. 3C).

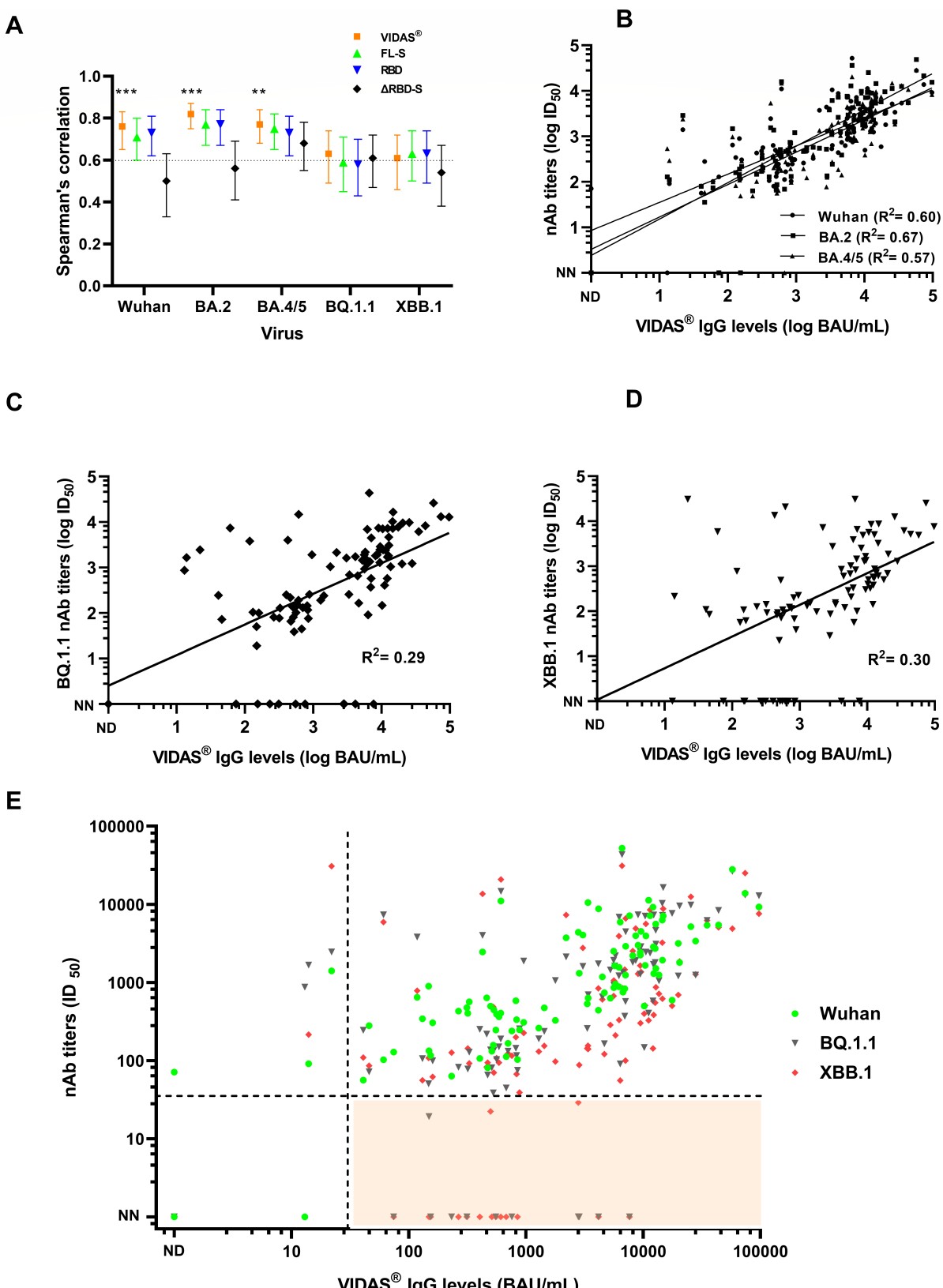

**FIG 2** Correlation between serological tests and neutralization assays. (A) Spearman's correlation between serological tests and neutralization assay against the Wuhan, BA.2, BA4/5, BQ1.1, and XBB.1 viruses. Spearman's correlations obtained with the VIDAS assay were compared with the $r_s$ value of 0.6 (horizontal line), which is considered a moderate correlation. Correlation values are shown with a 95% confidence interval. ***$P < 0.001$; **$P < 0.01$. (B) Agreement between VIDAS (Continued on next page)

Fig 2 (Continued)

IgG levels and neutralizing antibody titers against the Wuhan, BA.2, and BA4/5 viruses. The linear regression equations used were for Wuhan, $y = 0.6196x + 0.9300$ ($R^2 = 0.60$); BA.2, $y = 0.7115x + 0.5165$ ($R^2 = 0.67$); and BA.4/5, $y = 0.7115x + 0.5165$ ($R^2 = 0.57$). (C) Agreement between VIDAS IgG levels and neutralizing antibody titers against the BQ.1.1 virus; $y = 0.6700x + 0.4025$ ($R^2 = 0.29$). (D) Agreement between VIDAS IgG levels and neutralizing antibody titers against the XBB.1 virus; $y = 0.7037x + 0.03$ ($R^2 = 0.30$). (E) Comparison of neutralizing activities against the Wuhan, BQ.1.1, and XBB.1 viruses according to the VIDAS IgG levels. The orange zone highlights positive VIDAS sera (supplier positive threshold of 25 BAU/mL) without neutralizing antibody (titer < 30) against the BQ.1.1 and XBB.1 variants. nAb, neutralizing antibody; NN, non-neutralizing antibody; and ND, not detectable.

Because Omicron variants accumulate many mutations in the RBD, we also evaluated the in-house serological test based on the non-RBD spike. We speculated that this test would be more antigenically conserved relative to RBD and thus more suitable for neutralizing antibody detection in the Omicron era. A titer threshold of 26.9 and 84.5 optimized for the detection of neutralizing antibody against BQ.1.1 and XBB.1, respectively, using the Youden index was associated with a sensitivity and specificity of 0.83 (95% CI 0.73–0.90) and 0.85 (0.55–0.98) against BQ.1.1 and 0.73 (0.62–0.82) and 0.89 (0.65–0.99) against XBB.1 (Fig. 3B). The agreement between ΔRBD-S ELISA and neutralization assays was higher for BQ.1.1 ($κ = 0.47$, 95% CI 0.27–0.68) and XBB.1 ($κ = 0.44$, 95% CI 0.26–0.61) viruses than for Wuhan ($κ = 0.21$, 95% CI 0–0.46) or BA.4/5 ($κ = 0.19$, 95% CI 0.02–0.35) (Fig. 3D; Table S3). Compared with VIDAS serology, the ΔRBD-S ELISA had a better agreement with neutralization against the BQ.1.1 virus, although it was less adapted to detecting neutralizing antibodies against the Wuhan virus (Fig. 3D; Table S3).

## DISCUSSION

The measurement of anti-SARS-CoV-2-binding antibodies has been proposed as a correlate of protection in vaccinated individuals because of its association with neutralizing antibody levels (6–8). In the Omicron era, however, as new viruses have evaded the immunity induced by previous infections and vaccination, the correlation between binding antibody levels determined by serological assays developed from ancestral antigens and neutralizing antibody titers against new viruses needs to be re-examined. Here, we showed that the performance of binding antibody assays in predicting neutralizing activities of patient sera decreased with the successive Omicron waves.

We analyzed patterns of neutralization and cross-reactivity among a panel of five SARS-CoV-2 variants (Wuhan, BA.2, BA.4/5, BQ.1.1, and XBB.1) and five groups of human sera obtained from individuals following primary infection with the Wuhan virus in 2020 or infected with different Omicron variants, namely BA.2, BA.4/5, BQ.1.1, and XBB, between April 2022 and August 2023 in France. In contrast to the Wuhan virus-infected patients from 2020, the groups of patients exposed to the Omicron variants were likely more heterogeneous, reflecting the complexity of the population's immunity at the time of the study due to hybrid immunity, i.e., pre-existing immunity from vaccination and potentially previous multiple exposures to different SARS-CoV-2 variants (22). As expected, these patients had higher neutralizing antibody titers than the group of infected patients from 2020, who had only received a single antigenic stimulation by the ancestral virus. However, the sera of patients infected in 2020 were collected 6 months after infection, which may also have influenced their neutralizing antibody titers given the potential progressive loss of antibodies over time (12). Comparing the titers of neutralizing antibodies relative to the BA.2 virus, we confirmed that the successive Omicron waves were less sensitive to neutralizing antibodies. Consistent with previous reports, the BQ.1.1 and XBB.1 viruses were the least sensitive to neutralizing antibodies induced by prior infection or vaccination (23–26). By contrast, binding IgG titers obtained from serological tests based on spike or RBD ancestral antigen did not vary according to the subgroups of patients infected during the successive Omicron waves (BA.2 to XBB wave). The three binding antibody assays, based on spike and RBD antigens, showed similar performances in the detection of anti-SARS-CoV-2 IgGs and correlated well with neutralization against Wuhan, BA.2, and BA.4/5 viruses ($r_s$ ranging from 0.71 to 0.82). However, the correlations were lower for the BQ.1.1 and XBB.1 viruses (0.54–0.63). In

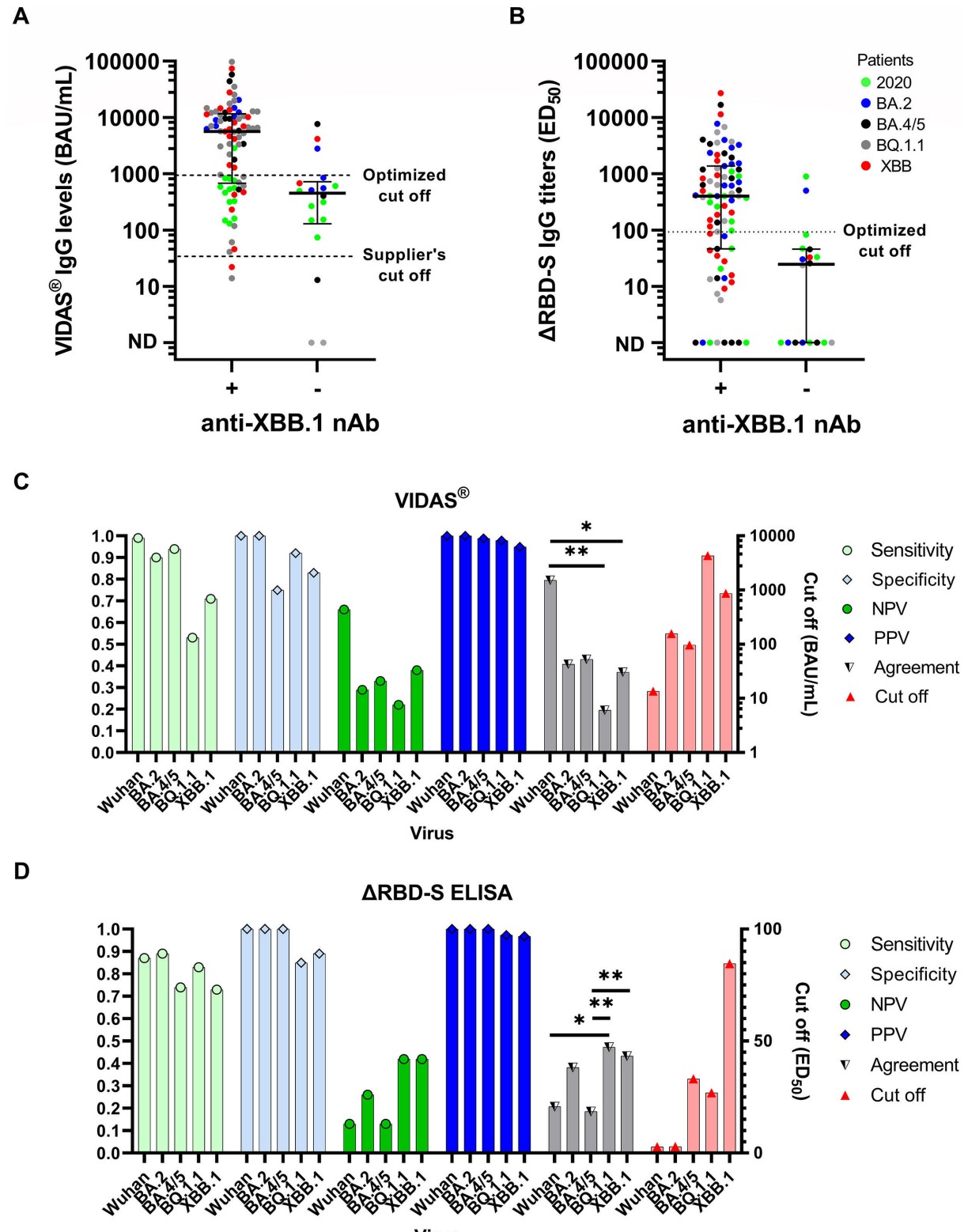

**FIG 3** Performance of serological tests at predicting neutralizing activities. (A) VIDAS IgG levels according to the presence (+) or absence (−) of neutralizing antibodies against the XBB.1 virus. (B) In-house ΔRBD-S ELISA IgG titers according to the presence (+) or absence (−) of neutralizing antibodies against the XBB.1 virus. The horizontal lines represent the median antibody titer values with the interquartile range. ND, not detectable. Green circles, 2020 patients; blue circles, BA.2 patients; black circles, BA.4/5 patients; gray circles, BQ1.1 patients; red circles, XBB patients. Performance of VIDAS assay (C) and ΔRBD-S ELISA (D) at predicting neutralizing activities against the Wuhan, BA.2, BA.4/5, BQ1.1, and XBB.1 viruses. The agreement was estimated using Cohen's Kappa coefficient. The

**Fig 3 (Continued)**

cutoff points were optimized using the Youden index and the measure of the receiver operating characteristic curve. NPV, negative predictive value; PPV, positive predictive value. **$P < 0.01$ and *$P < 0.05$ (for clarity, 95% confidence interval values for panels C and D are shown in Table S3).

addition, serology gave positive results with high IgG titers, even in the absence of detectable neutralizing antibody levels against the BQ.1.1 and XBB.1 viruses in patient sera (observed in 11% and 15% of patients, respectively, for the VIDAS assay). A formal comparison with the results of previous studies showing a weaker correlation between commercial serological and neutralization assays against the BA.1 and BA.2 Omicron variants remains difficult due to differences in the immune status of the populations under investigation (15, 17, 27). We showed that the ELISA based on a non-RBD spike was less sensitive in detecting binding IgG antibodies and weakly correlated with the neutralization assay against the Wuhan virus, which confirms that RBD was a major target of neutralizing antibodies in the pre-Omicron era. However, compared with other ELISAs, we did not observe a loss of correlation with the BQ.1.1 and XBB.1 viruses, suggesting that a test that does not include the RBD hotspot mutations may be better suited to detecting neutralizing antibody responses regardless of the virus' constant antigenic evolution (28, 29).

In line with recent reports showing that high levels of protection against symptomatic COVID-19 can be achieved at low serum neutralizing titers, we investigated levels of binding antibodies that could predict neutralizing antibody titers above 1:30 (8, 18). By optimizing the threshold of the prototype commercial test (VIDAS), the specificity for detecting neutralizing antibodies to the BQ.1.1 and XBB.1 viruses increased dramatically (five- to sixfold). Overall, we showed that the positivity thresholds of serological assays should be increased (4- to 173-fold depending on the virus) to improve the detection specificity of neutralizing antibodies against Omicron variants. Binding antibody levels, which have been associated with inverse risk of severe COVID-19, have also been used to optimize the timing of booster vaccinations (30, 31). Different binding antibody levels have been proposed as correlates of protection, ranging from moderate to high levels (140–7,000 BAU/mL), depending on the population or epidemic period under investigation (7, 8, 32–34). A threshold of 260 BAU/mL has been adopted by the French health authorities to classify patients as responders or non-responders to COVID-19 vaccines or to select patients for treatment with monoclonal antibody therapies based on a study evaluating the correlates of protection against the Alpha B.1.1.7 strain (7). Our results demonstrate that this cutoff is no longer appropriate for Omicron viruses, and an approach based on a new and higher threshold has not been established to date. When effective monoclonal antibody therapies against new variants become available, tests can be used to determine which patients should be treated. A recent report from the Infectious Disease Society of America recommends the use of spike serology to identify immunocompromised patients eligible for immune therapy when the result is negative. According to our study, if this strategy was adopted, no patients with positive serology would benefit from treatment, even though our findings show that a significant number of these patients have no neutralizing antibodies and are therefore potentially at risk of infection (15% of patients without neutralizing antibodies against XBB.1 virus had positive serology based on the supplier's cutoff). The use of specific optimized thresholds could improve the agreement between serology and neutralization against Omicron variants, including the BQ.1.1 and XBB.1 viruses. However, choosing an optimal threshold for clinical use is challenging. Indeed, we showed that a high and specific threshold would better identify patients who are candidates for immune therapy (despite missing three individuals), although the resulting lower sensitivity would lead to many patients being unnecessarily treated (29 individuals). The choice of a threshold value may thus depend on the risks and benefits of treatment, its availability, its costs, and the need to optimize the available resources.

Our study has several limitations. Our results are limited to the study of the Omicron variants of BA.2 to XBB and thus exclude the most recent viruses such as BA.2.86,

JN.1, and its subvariants, which are even more resistant to the immunity induced by vaccination and previous infections (35–39). Recent studies showed that sera from patients infected during the XBB.1.5 epidemic wave have similar neutralizing capacities against XBB.1.5, BA.2.86, and JN.1 viruses. However, these same sera are less effective at neutralizing JN.1 subvariants (KP.2, KP.3, KP2.3, and KP.3.1.1), with escape rates varying by a factor of 2–3 depending on the subvariant (37–39). The KP.3.1.1 variant, currently the most prevalent in France, has the highest capacity to evade neutralization of serum from patients infected or vaccinated with the XBB.1.5 virus (39). By applying these data to our study, we can hypothesize that the correlation between neutralization against the KP.3.1.1 variant and serology based on ancestral antigens is even weaker than that found for the XBB.1 virus. To achieve a stronger correlation between neutralization of JN.1 subvariants (KP.3.1.1) and serology, the binding antibody threshold determined by serology should probably be increased to a level that is even higher than that found for the XBB.1 virus in our study. It would certainly be optimal for serological tests to be antigenically matched to the currently circulating variants or the updated annual vaccine composition. However, this solution seems unlikely to be implemented in practice due to the regulatory constraints of the *in vitro* diagnostics market and the lack of a clear consensus on the clinical utility of serology, except in specific cases of immunocompromised patients (40). Given the variety of serological or neutralization assays that are currently available on the market, it will be challenging to establish a universal protective threshold or correlate that can be employed in all laboratories, despite the use of international standards (41–43). Each specialized center should propose a threshold of binding antibodies that correlates with the presence of neutralizing antibodies, with the aim of estimating a level of protection that would be useful for immunocompromised patients eligible for effective prophylactic treatment with monoclonal antibodies.

In conclusion, binding antibody levels do not necessarily reflect the level of protection against infection caused by new escaped variants. Serological tests that are adapted to the evolution of RBD or spike antigens would improve the correlation with neutralizing antibodies, as well as the clinical utility of serological testing in immunocompromised patients in the Omicron era.

## ACKNOWLEDGMENTS

We thank David Nemazee for providing the BQ.1.1 and XBB.1 plasmids (materials deposited at Addgene). We acknowledge access to the platforms of the Grenoble Instruct-ERIC center (IBS and ISBG; UMS 3518 CNRS-CEA-UGA-EMBL) within the Grenoble Partnership for Structural Biology (PSB), with support from FRISBI (ANR-10-INBS-05-02) and GRAL, a project of the University Grenoble Alpes graduate school (Ecoles Universitaires de Recherche) CBH-EUR-GS (ANR-17-EURE-0003). The IBS acknowledges integration into the Interdisciplinary Research Institute of Grenoble (IRIG, CEA) and financial support from CEA, CNRS, and UGA.

Léa Dépéry, Data curation, Formal analysis, Writing – review and editing | Isabelle Bally, Data curation, Formal analysis, Writing – review and editing | Axelle Amen, Data curation, Formal analysis, Writing – review and editing | Benjamin Némoz, Formal analysis, Software, Writing – review and editing | Marlyse Buisson, Formal analysis, Validation | Laurence Grossi, Validation | Aurélie Truffot, Writing – review and editing | Raphaële Germi, Writing – review and editing | Delphine Guilligay, Writing – review and editing | Mélanie Veloso, Data curation, Software, Writing – review and editing | Antoine Vilotitch, Data curation, Software, Writing – review and editing | Olivier Epaulard, Data curation, Writing – review and editing | Patrice Morand, Writing – review and editing | Winfried Weissenhorn, Resources, Writing – review and editing | Pascal Poignard, Conceptualization, Resources, Supervision, Validation, Writing – review and editing | Julien Lupo, Conceptualization, Data curation, Formal analysis, Funding acquisition, Investigation, Methodology, Project administration, Resources, Supervision, Validation, Writing – original draft, Writing – review and editing.

## AUTHOR AFFILIATIONS

[1]Institut de Biologie Structurale (IBS), CEA, CNRS, Université Grenoble Alpes, Grenoble, France

[2]Laboratoire d'Immunologie, CHU Grenoble Alpes, Université Grenoble Alpes, Grenoble, France

[3]Laboratoire de Virologie, CHU Grenoble Alpes, Université Grenoble Alpes, Grenoble, France

[4]Cellule d'ingénierie des données, Département de Santé Publique, CHU Grenoble Alpes, Université Grenoble Alpes, Grenoble, France

[5]Service des Maladies Infectieuses, CHU Grenoble Alpes, Université Grenoble Alpes, Grenoble, France

## AUTHOR ORCIDs

Olivier Epaulard http://orcid.org/0000-0001-8282-9522
Julien Lupo http://orcid.org/0000-0002-6755-3115

## FUNDING

| Funder | Grant(s) | Author(s) |
| --- | --- | --- |
| bioMérieux (bioMérieux SA) | | Julien Lupo |

## AUTHOR CONTRIBUTIONS

Léa Dépéry, Conceptualization, Data curation, Formal analysis, Funding acquisition, Investigation, Methodology, Project administration, Resources, Supervision, Validation, Writing – original draft, Writing – review and editing | Isabelle Bally, Conceptualization, Data curation, Formal analysis, Resources, Writing – review and editing | Axelle Amen, Conceptualization, Data curation, Formal analysis, Resources, Writing – review and editing | Benjamin Némoz, Data curation, Formal analysis, Resources, Software, Writing – review and editing | Marlyse Buisson, Conceptualization, Formal analysis, Resources, Software, Validation | Laurence Grossi, Conceptualization, Resources, Validation | Aurélie Truffot, Resources, Writing – review and editing | Raphaële Germi, Writing – review and editing | Delphine Guilligay, Resources, Writing – review and editing | Mélanie Veloso, Data curation, Resources, Software, Writing – review and editing | Antoine Vilotitch, Data curation, Software, Writing – review and editing | Olivier Epaulard, Data curation, Software, Writing – review and editing | Patrice Morand, Writing – review and editing | Winfried Weissenhorn, Resources, Writing – review and editing | Pascal Poignard, Conceptualization, Resources, Supervision, Validation, Writing – review and editing | Julien Lupo, Conceptualization, Data curation, Formal analysis, Funding acquisition, Investigation, Methodology, Project administration, Resources, Supervision, Validation, Writing – original draft, Writing – review and editing

## ETHICS APPROVAL

The study was approved and assigned reference number 38RC23.0067 by the Directorate for Scientific Research at the University Hospital Grenoble Alpes.

## ADDITIONAL FILES

The following material is available online.

### Supplemental Material

**Figure S1 (Spectrum01568-24-s0001.docx).** Neutralizing antibodies titers against the Wuhan, BA.2, BA.4/5, BQ1.1, and XBB.1 viruses in patients infected in 2020 and during the successive Omicron waves.

**Supplemental tables (Spectrum01568-24-s0002.docx).** Tables S1 to S3.

Open Peer Review

**PEER REVIEW HISTORY (review-history.pdf).** An accounting of the reviewer comments and feedback.

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
