## [Reviewer comments · Microbiology Spectrum]

Microbiology Spectrum

Anti-SARS-CoV-2 serology based on ancestral RBD antigens does not correlate with the presence of neutralizing antibodies against Omicron variants

Léa Dépéry, Isabelle Bally, Axelle Amen, Benjamin Nemoz, Marlyse Buisson, Laurence Grossi, Aurélie Truffot, Raphaele Germe, Delphine Guilligay, Mélanie Veloso, Antoine Vilotitch, Olivier Epaulard, Patrice Morand, Winfried Weissenhorn, Pascal Poignard, and Julien Lupo

Corresponding Author(s): Julien Lupo, Centre Hospitalier Universitaire Grenoble Alpes

Review Timeline:

Submission Date:	June 27, 2024
Editorial Decision:	September 3, 2024
Revision Received:	October 14, 2024
Accepted:	October 26, 2024

Editor: Chuan Lim

Reviewer(s): The reviewers have opted to remain anonymous.

Transaction Report:

DOI: <https://doi.org/10.1128/spectrum.01568-24>

Re: Spectrum01568-24 (**Utility of anti-SARS-CoV-2 serologic testing to predict the presence of neutralizing antibodies against Omicron variants**)

Dear Dr. Julien LUPO:

Thank you for the privilege of reviewing your work. Below you will find my comments, instructions from the Spectrum editorial office, and the reviewer comments.

Revision Guidelines

Sincerely,
Chuan Lim
Editor
Microbiology Spectrum

Reviewer #1 (Comments for the Author):

In this study, Dépéry et al examined the relationship between antibody binding and antibody neutralization for sera collected from 100 patients infected with different SARS-CoV-2 variants. The authors found that binding and neutralization correlated well for earlier Omicron variants, but later variants such as BQ.1.1 and XBB had weak correlation, suggesting that serological tests based on ancestral spike/RBD may not be predictive of neutralizing capacity against contemporary strains.

The experimental methodology is appropriate and the manuscript is written well. The results overlap in part with other prior reports, but a comprehensive systematic analysis is welcomed.

My main comment would be that it is not clear that the cohorts are directly comparable, and that the results should therefore be focused on within-group comparisons, such as those shown in Figure S1, rather than the aggregated results shown in the main figures. This concern rises out of the following: (1) individuals infected by the Wuhan variant were unvaccinated, indicating that they only had one exposure, whereas the Omicron variant group has had many exposures through vaccination/infection and varying levels of immune imprinting, (2) the time post-infection is significantly longer for the Wuhan group, and it is well recognized the rapid decay of SARS-CoV-2 antibodies, and (3) the Omicron group contains a large proportion of patients that would be expected to mount a poor response (e.g., dialysis patients, transplant patients). It is also not clear if the distribution of these non-hospitalized patients is equal across the four Omicron groups.

Minor comments:

- "BQ.1.1" is missing a period and written as "BQ1.1" throughout the manuscript and figures.
- It is hard to see the lines representing means for the BA.4/5 group given the overlap in color.

Reviewer #2 (Comments for the Author):

The authors explore the correlation between binding antibody levels determined by serological assays developed from ancestral antigens and neutralizing antibody titers against new SARS-CoV-2 variants. The question is of major importance for immunocompromised patients that could develop severe respiratory or sepsis infections in cases of neutralizing antibody default and who will be tested by classical serological assays in 2024. Moreover, the authors confirmed by their investigations the low neutralizing effects of ancient anti-SARS-Cov-2 antibodies against the new emerging viral forms with specific Spike epitopes. However, in the present manuscript version, the authors did not test Ab neutralization assays against the newly emerging variants as JN.1 and its sub-lineages. At the present time, Health authorities such as the FDA (Food and Drug Administration) and the EMA (European Medicines Agency) are recommending that JN.1 vaccines be adapted for the 2024/2025 vaccination campaigns in order to better target circulating variants. Therefore, some additional experiments (major concerns) have to be performed to up-date and to improve the present manuscript and the utility of commercially available serological tests used in clinical practice.

Minor concerns:

- The title is not informative about the obtained data and could be improved as followed; Absence of correlation between binding antibody levels determined by serological assays developed from ancestral antigens and neutralizing antibody titers against new SARS-CoV-2 variants (or a more concise title can be obtained ...)
- The first minor question concerns the validation of the neutralization assays using pseudotyped viruses and its correlation with the classical BSL3 viral culture approach using referenced batches or QCMDs of neutralizing Abs (Nature 600, 701-706 (2021)).
- The second minor point concerns the biostatistical analyses and the generation of the figures; have they been generated or validated by a real biostatistical expert? Who is he ?
- The third minor point is the absence of ethical review agreement number issued from a referenced French ethical committee (lines 119-121).
- The four minor point is a question; have the authors evidenced statistical differences between sera samples before and after the vaccination era for each tested subgroup of patients.

Major concerns:

- One of my major concerns is that the authors did not include new genetic circulating variants. Therefore, I strongly recommend to test JN.1 variants and its sub lineages (specifically KP3.1.1) using their pseudo typed SARS-Cov-2 technical strategy against well selected 30 to 35 ancient serum samples. This approach will allow to up-date the present study and results commercial serological tests but also it will provide new data on the immune capacities of SARS-CoV-2 infected or vaccinated patients to fight against severe infection by these variants. The obtained data will be valuable to discuss the protection against SARS-CoV-2 severe infections in immunocompromised patients.
- What about the next classical serological assays or the upgrade of further commercially available serological tests? The authors have to specifically discuss this point.
- Lines 384 -387: the authors claim that given the heterogeneous results for tests measuring binding antibodies and the variability of pre-existing immunity from one individual to another the thresholds obtained in their study cannot be universally adopted in routine clinical practice. What's about real solutions or improvements could be proposed? universal QCMDS, new serological tests, new ab neutralizing Ab assays? others,please discuss this point!!

The authors explore the correlation between binding antibody levels determined by serological assays developed from ancestral antigens and neutralizing antibody titers against new SARS-CoV-2 variants. The question is of major importance for immunocompromised patients that could develop severe respiratory or sepsis infections in cases of neutralizing antibody default and who will be tested by classical serological assays in 2024. Moreover, the authors confirmed by their investigations the low neutralizing effects of ancient anti-SARS-Cov-2 antibodies against the new emerging viral forms with specific Spike epitopes. However, in the present manuscript version, the authors did not test Ab neutralization assays against the newly emerging variants as JN.1 and its sub-lineages. At the present time, Health authorities such as the FDA (Food and Drug Administration) and the EMA (European Medicines Agency) are recommending that JN.1 vaccines be adapted for the 2024/2025 vaccination campaigns in order to better target circulating variants. Therefore, some additional experiments (major concerns) have to be performed to up-date and to improve the present manuscript and the utility of commercially available serological tests used in clinical practice.

Minor concerns:

- The title is not informative about the obtained data and could be improved as followed; Absence of correlation between binding antibody levels determined by serological assays developed from ancestral antigens and neutralizing antibody titers against new SARS-CoV-2 variants (or a more concise title can be obtained ...)
- The first minor question concerns the validation of the neutralization assays using pseudotyped viruses and its correlation with the classical BSL3 viral culture approach using referenced batches or QCMDs of neutralizing Abs (*Nature* **600**, 701–706 (2021)).
- The second minor point concerns the biostatistical analyses and the generation of the figures; have they been generated or validated by a real biostatistical expert? Who is he ?
- The third minor point is the absence of ethical review agreement number issued from a referenced French ethical committee (lines 119-121).
- The four minor point is a question; have the authors evidenced statistical differences between sera samples before and after the vaccination era for each tested subgroup of patients.

Major concerns:

- One of my major concerns is that the authors did not include new genetic circulating variants. Therefore, I strongly recommend to test JN.1 variants and its sub lineages (specifically KP3.1.1) using their pseudo typed SARS-Cov-2 technical strategy against well selected 30 to 35 ancient serum samples. This approach will allow to up-date the present study and results commercial serological tests but also it will provide new data on the immune capacities of SARS-CoV-2 infected or vaccinated patients to fight against severe infection by these variants. The obtained data will be valuable to discuss the protection against SARS-CoV-2 severe infections in immunocompromised patients.
- What about the next classical serological assays or the upgrade of further commercially available serological tests? The authors have to specifically discuss this point.
- Lines 384 -387: the authors claim that given the heterogeneous results for tests measuring binding antibodies and the variability of pre-existing immunity from one individual to another the thresholds obtained in their study cannot be universally adopted in routine clinical practice. What's about real solutions or improvements

could be proposed? universal QCMDS, new serological tests, new ab neutralizing Ab assays? others,please discuss this point!!

Response to the Reviewers

Manuscript ID: **Spectrum01568-24**

Title (new): **Anti-SARS-CoV-2 serology based on ancestral RBD antigens does not correlate with the presence of neutralizing antibodies against Omicron variants**

Authors : Léa Dépéry, Isabelle Bally, Axelle Amen, Benjamin Némoz, Marlyse Buisson, Laurence Grossi, Aurélie Truffot, Raphaële Germe, Delphine Guilligay, Mélanie Veloso, Antoine Vilotitch, Olivier Epaulard, Patrice Morand, Winfried Weissenhorn, Pascal Poignard and Julien Lupo

We thank the Reviewers for their constructive comments and suggestions, which allowed us to improve the quality of the manuscript. The Reviewers' comments were carefully addressed, as detailed below. Additions and changes made in the text can be viewed in the co-submitted marked version of the revised manuscript (named « Manuscript_marked »). Please note that the page and line numbers listed below refer to the marked version of the revised manuscript.

Reviewer #1 (Comments for the Author):

In this study, Dépéry et al examined the relationship between antibody binding and antibody neutralization for sera collected from 100 patients infected with different SARS-CoV-2 variants. The authors found that binding and neutralization correlated well for earlier Omicron variants, but later variants such as BQ.1.1 and XBB had weak correlation, suggesting that serological tests based on ancestral spike/RBD may not be predictive of neutralizing capacity against contemporary strains. The experimental methodology is appropriate and the manuscript is written well. The results overlap in part with other prior reports, but a comprehensive systematic analysis is welcomed.

My main comment would be that it is not clear that the cohorts are directly comparable, and that the results should therefore be focused on within-group comparisons, such as those shown in Figure S1, rather than the aggregated results shown in the main figures. This concern rises out of the following: (1) individuals infected by the Wuhan variant were unvaccinated, indicating that they only had one exposure, whereas the Omicron variant group has had many exposures through vaccination/infection and varying levels of immune imprinting, (2) the time post-infection is significantly longer for the Wuhan group, and it is well recognized the rapid decay of SARS-CoV-2 antibodies, and (3) the Omicron group contains a large proportion of patients that would be expected to mount a poor response (e.g., dialysis patients, transplant patients). It is also not clear if the distribution of these non-hospitalized patients is equal across the four Omicron groups.

Response:

We thank the Reviewer for its relevant comments.

Our study population was deliberately heterogeneous, consisting of patients infected at different times during the pandemic. This reflected the overall and varied immune status of the population at the time of our work. We agree with the reviewer that patients infected in 2020 represent a distinct group that differs from the other Omicron subgroups in terms of antibody characterization (no vaccination, 6-month serum collection period, no multiple exposure) and that this should be discussed in our article.

We believe it would be beneficial to retain these patients as a point of reference. The aim of our manuscript was not to compare patient subgroups with one another, but rather to compare the antibody reactivity of a heterogeneous patient population to different pseudoviruses or antigens in different ELISAs. The results have been revised by removing the data belonging to the 2020 patient subgroup, yet the conclusions remain unchanged (see the Table above, Omicron patients n=80). The first part of our results (Figure 1A) is a global characterization of neutralizing antibody titers against the Wuhan strain and the different Omicron variants in our population. We observed lower neutralizing antibody titers against the Omicron variants that were the most recent circulating viruses at the time of our study (BQ.1.1 and XBB). A sub-group analysis would have been interesting (as shown in Figure S1), but the relatively small size of our populations limited our statistical power. Our results are presented in aggregate form to reflect the diversity of the immune status of a population at a given time. The primary objective of our study is not to compare the ability of different group of patients to neutralize omicron viruses but to assess the correlation between binding and neutralizing antibodies.

Spearman's correlation between serology and neutralization assays against Wuhan, BA.2, BA4/5, BQ.1.1, and XBB.1 viruses for Omicron patients (n=80):

Spearman's correlation [CI 95%]	Spike ELISA	RBD ELISA	Spike-ΔRBD ELISA	VIDAS® assay
PV Wuhan	0.64 [0.48 - 0.75]	0.65 [0,50 - 0,76]	0.44 [0,24 - 0,60]	0.69 [0,55 - 0,79]
PV BA.2	0.70 [0.56 - 0.80]	0.69 [0,54 - 0,79]	0.52 [0,33 - 0,66]	0.75 [0,63 - 0,83]
PV BA.4/5	0.72 [0.59 - 0.81]	0.67 [0,53 - 0,79]	0.68 [0,54 - 0,79]	0.73 [0,61 - 0,82]
PV BQ.1.1	0.51 [0.32 - 0.66]	0.48 [0,28 - 0,63]	0.60 [0,43 - 0,73]	0.57 [0,39 - 0,70]
PV XBB.1	0.56 [0.38 - 0.70]	0.52 [0,33 - 0,66]	0.51 [0,32 - 0,66]	0.51 [0,32 - 0,66]

Specific responses:

(1) (2) We would like to clarify that the sera of patients infected in 2020 were collected six months after infection. This also influenced their neutralizing antibody titers (in addition to a single antigenic stimulation), given the progressive loss of antibodies over time (see modifications, lanes 329-331).

(3) Due to the retrospective nature of our study, we selected patients with PCR and sequencing results who could benefit from serum sampling in our laboratory. Patients who met these criteria were primarily those with chronic medical conditions that are regularly monitored in the hospital setting. As recommended, we have updated Table S1 to include more detailed information on the patient subgroups (due to the limited sample size, statistical comparisons were not feasible).

Table S1 (modified in the new version):

	Wuhan n=20	BA.2 n=20	BA.4/5 n=20	BQ.1.1 n=20	XBB n=20
Sex, male (%)	70	45	60	70	45
Median age [IQR]	67 [58.25-70.75]	77 [62-86.5]	67 [53.25-82.75]	74 [63.5-83.25]	74.5 [49.25-85]
Median delay between collection and PCR diagnosis, day [IQR]	170 [165-190]	36 [26-42]	38.5 [25.25-51.25]	34 [26-40]	24 [22-27]
Strain sequencing yes/No (No.)	No	yes	yes	yes	Yes XBB (3) XBB.1.5 (9) XBB.1.9 (8)
Vaccination with two or more doses (BNT 162b2 or mRNA1273 or chAdOx1 + BNT 162b2/mRNA1273) Adapted bivalent vaccines (including Omicron BA4/5 valence) were administered from autumn 2022 (for the fourth or more dose) (No.)	0	16	17	16	12
Vaccination not documented (No.)	NA	4	3	4	6
Origin of the patients					
Non-hospitalized patients (n=62):	0	16	14	17	15
Dialysis patients (n=35)		9	8	11	7
Transplant patients (n=17)		5	5	3	4
Gynecology consultation (n=3)		1	1	0	1
Infectious disease consultation (n=2)		0	0	1	1
Health care professionals (n=5)		1	0	2	2
Hospitalized patients (n=18)	20	4	6	3	5

Minor comments:

- "BQ.1.1" is missing a period and written as "BQ1.1" throughout the manuscript and figures.
- It is hard to see the lines representing means for the BA.4/5 group given the overlap in color.

Response:

The error in BQ.1.1 has been corrected throughout the manuscript, figures and tables. Figure S1 has been modified to improve the visibility of the median antibody titers for the BA.4/5 group.

Reviewer #2 (Comments for the Author):

The authors explore the correlation between binding antibody levels determined by serological assays developed from ancestral antigens and neutralizing antibody titers against new SARS-CoV-2 variants. The question is of major importance for immunocompromised patients that could develop severe respiratory or sepsis infections in cases of neutralizing antibody default and who will be tested by classical serological assays in 2024. Moreover, the authors confirmed by their investigations the low neutralizing effects of ancient anti-SARS-Cov-2 antibodies against the new emerging viral forms with specific Spike epitopes. However, in the present manuscript version, the authors did not test Ab neutralization assays against the newly emerging variants as JN.1 and its sub-lineages. At the present time, Health authorities such as the FDA (Food and Drug Administration) and the EMA (European

Medicines Agency) are recommending that JN.1 vaccines be adapted for the 2024/2025 vaccination campaigns in order to better target circulating variants. Therefore, some additional experiments (major concerns) have to be performed to up-date and to improve the present manuscript and the utility of commercially available serological tests used in clinical practice.

Minor concerns:

- The title is not informative about the obtained data and could be improved as followed; Absence of correlation between binding antibody levels determined by serological assays developed from ancestral antigens and neutralizing antibody titers against new SARS-CoV-2 variants (or a more concise title can be obtained ...)

Response:

As suggested, we modified the title of our paper as follows: “Anti-SARS-CoV-2 serology based on ancestral RBD antigens does not correlate with the presence of neutralizing antibodies against Omicron variants”

- The first minor question concerns the validation of the neutralization assays using pseudotyped viruses and its correlation with the classical BSL3 viral culture approach using referenced batches or QCMDs of neutralizing Abs (Nature 600, 701-706 (2021)).

Response:

We use several quality criteria to validate a neutralization plate (96 wells) and antibody controls rather than molecular or viral controls to validate our experiments. Firstly, we check that the light intensity exceeds 150,000 RLU in the wells without sera (0% neutralization, raw H) to verify that the intensity signal is well above the background noise signal (PV were previously tittered using HeLa ACE-2 to determine the appropriate dilution of PV necessary to obtain about 200,000 RLU per well). The first column contains no virus (blank). The last column (position 12) corresponds to the dilution of the antibody or serum used as a positive control (mAb or patient's serum, depending on the PV to be neutralized). We check that the ID50 corresponding to the positive control is reproducible between assays and consistent with expected values (values compared with our previous experiments and works or the literature if available for mAb). The same approach was employed for ELISAs. For illustrative purposes, we present a neutralization plate result for PV BQ.1.1 with sera from patients infected in 2020 as an example. The complete set of raw data is available upon request.

Method name: HIV Corning half u clear
 Application: SparkControl V2.1
 Device: Spark 10M Serial number: 1701003044
 Firmware: LUM:V5.2.3|ABS:V4.3.1|ABS_MEX:V5.0.7|MTF:V12.4.0|FLUOR:V5.1.2

Date: 2023-06-15
 Time: 10:51
 System: PC-GDQQWG2
 User: PC-GDQQWG2/COMPUTER
 Plate: [COR96w half area clear bottom] - Corning 96 Flat White [COR96w half area clear bottom]
 Lid lifter: No lid
 Humidity Cassette: No humidity cassette
 Smooth mode: Not selected

List of actions in this measurement script:
 Plate
 Shaking
 Luminescence lumi

Name: COR96w half area clear bottom
 Plate layout: A1-H12
 Plate area: A1-H12

Start Time: 2023-06-15 10:50:41
 Shaking (Orbital) Duration: 5 s
 Shaking (Orbital) Position: Current
 Shaking (Orbital) Amplitude: 3 mm
 Shaking (Orbital) Frequency: 180 rpm
 End Time: 2023-06-15 10:50:46

Mode: Luminescence
 Name: lumi
 Attenuation: Automatic
 Attenuation color: ODO
 Settle time: 0 ms
 Integration time: 100 ms
 Output: Counts / s
 Part of Plate: A1-H12

Start Time: 2023-06-15 10:50:48
 Temperature: 23,8 °C

<>	1	2	3	4	5	6	7	8	9	10	11	12	
A	200	2690	2690	4491	16564	6062	13259	8414	2430	680	6993	8844	280
B	250	12208	17195	34492	18618	23158	19249	6412	7523	17125	22115	4701	
C	320	33187	42474	52453	36118	47949	37443	22576	18438	38929	32756	17746	
D	230	92573	87294	69178	65945	82361	58668	74639	40596	87708	96623	29124	
E	360	106395	103524	106355	85529	76160	73903	93704	78197	134129	99906	41580	
F	360	145746	142407	113243	93371	91968	83693	115338	106638	121928	137090	92856	
G	290	142661	141646	141585	122435	119873	104404	157276	136756	132345	144335	21870	
H	440	178736	154064	164831	166286	165421	168229	171598	156514	149625	174184	158679	

End Time: 2023-06-15 10:51:42

		1	2	3	4	5	6	7	8	9	10	11	12						
plaque 2 HeLa														P.contr	ole	RLU 100	RLU 100	RLU100	RLU NI
BO1														P.contr	ole	RLU 100	RLU 100	RLU100	RLU NI
serum	NI	WTP11	WTP12	WTP13	WTP14	WTP15	WTP16	WTP17	WTP18	WTP19	WTP20	P.contr	ole	RLU 100	RLU 100	RLU100	RLU NI		
A	30	2690	4491	16564	6062	13259	8414	2430	680	6993	8844	280	30	178736					
B	90	12208	17195	34492	18618	23158	19249	6412	7523	17125	22115	4701	90	154064					
C	270	33187	42474	52453	36118	47949	37443	22576	18438	38929	32756	17746	270	164831					
D	810	92573	87294	69178	65945	82361	58668	74639	40596	87708	96623	29124	810	166286					
E	2430	106395	103524	106355	85529	76160	73903	93704	78197	134129	99906	41580	2430	165421					
F	7290	145746	142407	113243	93371	91968	83693	115338	106638	121928	137090	92856	7290	168229					
G	21870	142661	141646	141585	122435	119873	104404	157276	136756	132345	118232	144335	21870	171598					
H	H réel	164948,8	164949	164949	164949	164949	164949	164949	164949	164949	164949	164949	164948,8	156514	149625				
														174184					
														158679					
BO1														P.contr	ole	RLU 100	RLU 100	RLU100	RLU NI
serum	NI	WTP11	WTP12	WTP13	WTP14	WTP15	WTP16	WTP17	WTP18	WTP19	WTP20	P.contr	ole	RLU 100	RLU 100	RLU100	RLU NI		
30		98,37	97,28	89,96	96,32	91,96	94,90	98,53	99,59	95,76	94,64	99,83	30						
90		92,60	89,58	79,09	88,71	85,96	88,33	96,11	95,44	89,62	86,59	97,15	90						
270		79,88	74,25	68,20	78,10	70,93	77,30	86,31	86,82	76,40	80,14	89,24	270						
810		43,88	47,08	38,06	50,02	50,07	54,43	54,75	75,39	46,83	41,42	82,34	810						
2430		35,50	37,24	35,32	48,15	53,83	55,20	43,19	52,59	18,68	39,43	74,79	2430						
7290		11,64	13,67	31,35	43,39	44,24	49,26	30,08	35,35	26,08	16,89	43,71	7290						
21870		13,51	14,13	14,16	25,77	27,33	36,71	4,65	17,09	19,77	28,32	12,50	21870						
		0,00	0,00	0,00	0,00	0,00	0,00	0,00	0,00	0,00	0,00	0,00							

In the Table above, we can see our inter-assay reproducibility (each plate or each experiment specific for a PV contained a positive control):

Positive control used in our experiments to assess the inter-assay reproducibility:

			PV WT	PV BA.2	PV BA.4/5	PV BQ.1.1	PV XBB.1	
IC50 Evusheld (ng/mL)	Patients BA.2	Plate 1	6.955	16.279				
		Plate 2	3.501	22.362				
	Patients BA.4/5	Plate 1	7.409	10.270	72.649			
		Plate 2	10.884	11.008	51.624			
IC50 Cilgavimab (ng/mL)	Patients 2020	Plate1		4.437	40.865			
		Plate 2		4.999	27.222			
	Patients BA.2	Plate 1			26.473			
		Plate 2			31.011			
	Patients BQ.1.1	Plate 1		6.016	30.467			
		Plate 2		3.929	26.270			
	Patients XBB	Plate 1		7.214	20.562			
		Plate 2		9,145	40.143			
IC50 CC12.1 (ng/mL)	Patients 2020	Plate 1	20.792					
		Plate 2	41.590					
	Patients BQ.1.1	Plate 1	50.121					
		Plate 2	43.212					
	Patients XBB	Plate 1	18.147					
		Plate 2	22.985					
ID50 patient control 1*	Patients 2020	Plate 1					7290	
		Plate 2					5234	
	Patients BA.2	Plate 1				7168	4127	
		Plate 2				4682	2891	
	Patients BA.4/5	Plate 1				2297	1720	
		Plate 2				2507	2346	
	Patients BQ.1.1	Plate 1				6528	4715	
		Plate 2				5117	5250	
	ID50 patient control 2*	Patients 2020	Plate 1				7294	
			Plate 2				6370	
Patients XBB		Plate 1				4853	5435	
		Plate 2				6120	5898	

*Control sera issued from patients infected during the BA.4/5 wave with high neutralizing antibody titers against BQ.1.1 and XBB.1 pseudoviruses.

- The second minor point concerns the biostatistical analyses and the generation of the figures; have they been generated or validated by a real biostatistical expert? Who is he ?

Response:

The statistical analyses were conducted by a team of biostatisticians (Dr. Antoine Vilotitch and Dr. Mélanie Veloso, who are the co-authors). A 50-page analysis report has been prepared and is available upon request. Figures and statistical analyses have been validated by their expertise (the figures and p-values were generated using GraphPad Prism version 9 or Stata version 15 software as appropriate).

- The third minor point is the absence of ethical review agreement number issued from a referenced French ethical committee (lines 119-121).

Response:

This non-interventional retrospective study involving data and samples from human participants has been carried out in CHU Grenoble Alpes according to French current regulation. The investigators have signed a commitment to comply with Reference Methodology n°004 issued by French Authorities (CNIL). Subjects were all informed and did not oppose; written consent was not required for the retrospective study in accordance with the national legislation and the institutional requirements. Our study was assigned reference number 38RC23.0067 by the scientific research department (Direction de la Recherche Scientifique) of the Grenoble hospital center. We have added this information to the method section (lines 123-127).

- The fourth minor point is a question; have the authors evidenced statistical differences between sera samples before and after the vaccination era for each tested subgroup of patients.

Response:

Unfortunately, this question could not be addressed due to the retrospective nature of the study. We did not have pre-vaccination sera for the majority of our patients. Our cohort included 100 different patients and not a cohort of 20 patients followed longitudinally between 2020 and 2023.

Major concerns:

- One of my major concerns is that the authors did not include new genetic circulating variants. Therefore, I strongly recommend to test JN.1 variants and its sub lineages (specifically KP3.1.1) using their pseudo typed SARS-CoV-2 technical strategy against well selected 30 to 35 ancient serum samples. This approach will allow to up-date the present study and results commercial serological tests but also it will provide new data on the immune capacities of SARS-CoV-2 infected or vaccinated patients to fight against severe infection by these variants. The obtained data will be valuable to discuss the protection against SARS-CoV-2 severe infections in immunocompromised patients.

Response:

We concur with the reviewer's assessment regarding the necessity of aligning virus testing with recent epidemiological trends, updating WHO recommendations for vaccine composition, and identifying clinically meaningful serological protection thresholds for immunocompromised patients. Unfortunately, despite the relevance of the request, we are unable to repeat the experiments with JN.1 variants and its sub-lineages due to a lack of available staff and funds to continue the project. Our experiments were conducted between January and August 2023 with the assistance of an M2 student and funding tailored to the scope of our work and the size of our study population. We tested the neutralization against the latest variants in circulation at the time of our work (XBB.1), but unfortunately, we were unable to publish our results in real time. We presented preliminary results at the RICAI congress in Paris in December 2023 and submitted the article to an ASM journal in May 2024. We recognize that this time lapse between results and publication may diminish the value of our work, particularly given the rapid evolution of viral strains during successive SARS-CoV-2 epidemics.

It is possible that in a few months' time new strains will emerge, and that new data obtained with today's strains will also become outdated. We believe that general lessons can be drawn from our study, despite the fact that our results are not up to date with the most recent variants at the time of publication. In response to this significant observation, we have conducted an exhaustive review of the literature on JN.1 viruses and their sublineages, with a particular emphasis on their capacity to evade immunity induced by prior infections or vaccinations. Recent studies have demonstrated that sera from patients infected during the XBB1.5 epidemic wave exhibit comparable neutralizing capacities against XBB1.5, BA.2.86, EG.5.1, and JN.1 viruses. However, these same sera demonstrated reduced efficacy in neutralizing JN.1 sublineages (KP.2, KP.3, KP2.3, KP.3.1.1), with escape rates varying by a factor of 2-3 depending on the sublineage. The KP.3.1.1 variant has the highest capacity to evade neutralization of serum from patients infected or vaccinated with the XBB.1.5 strain. Applying these data to our study, we can hypothesize that the correlation between neutralization against the KP.3.1.1 variant and serology based on ancestral antigens is even weaker than that observed for PV XBB.1. To achieve a greater degree of concordance between the neutralization of JN.1 sublineages (KP.3.1.1) and serological results, the binding antibody threshold determined by serology should likely be further increased compared to that identified for the XBB.1 virus in our study. A new paragraph has been added to discuss this point in further detail and three new references were added (**lanes 397-408, references 37,38 and 39**).

- What about the next classical serological assays or the upgrade of further commercially available serological tests? The authors have to specifically discuss this point.

Response:

We agree with the Reviewer. It is important to specify how serological tests could be adapted to better correlate with neutralization. We have added a specific paragraph to the discussion on the need to adapt serological tests to viral epidemiology, as is the case with vaccines, which are appropriately updated with the latest circulating strains. An alternative approach could be to develop a serological test with conserved antigens that could potentially remain unchanged as viral epidemiology evolves (see modifications, **lanes 410-414**, insertion of the **reference 40**).

- Lines 384 -387: the authors claim that given the heterogeneous results for tests measuring binding antibodies and the variability of pre-existing immunity from one individual to another the thresholds obtained in their study cannot be universally adopted in routine clinical practice. What's about real solutions or improvements could be proposed? universal QCMS, new serological tests, new ab neutralizing Ab assays? others,please discuss this point!!

Response:

We thank the reviewer for its relevant comment, in line with the previous ones. It would be optimal for each laboratory to validate the serological test it uses in routine practice to quantify anti-SARS-CoV-2 antibodies with a neutralization assay adapted to the current viral epidemiology. However, not all laboratories are able to implement a neutralization assay. Given the multiplicity of serological or neutralization assays currently on the market, it will be challenging to determine a universal protection threshold or correlate that can be used in all laboratories, despite the use of international standards. The use of an international standard is not always a matter of course, and inter-laboratory variability may persist (as in the case of anti-rubella antibodies, for example). Each specialized center could

validate and propose a threshold of protection for people who need it (immunocompromised patients eligible for effective treatment in order to save resources if they are limited). A new paragraph was added in the discussion to address this important question (**lanes 414-421**).

Re: Spectrum01568-24R1 (Anti-SARS-CoV-2 serology based on ancestral RBD antigens does not correlate with the presence of neutralizing antibodies against Omicron variants)

Dear Dr. Julien LUPO:

Your manuscript has been accepted, and I am forwarding it to the ASM production staff for publication. Your paper will first be checked to make sure all elements meet the technical requirements. ASM staff will contact you if anything needs to be revised before copyediting and production can begin. Otherwise, you will be notified when your proofs are ready to be viewed.

Sincerely,
Chuan Lim
Editor
Microbiology Spectrum

Reviewer #1 (Comments for the Author):

The authors have addressed my prior comments in this revision. A minor note that the last row in the updated Table S1, "Hospitalized patients", sums to 38 but is noted as "n=18".

Reviewer #2 (Comments for the Author):

The authors have taken into account all of the reviewer's remarks and queries. They have subsequently improved the proposed manuscript and they added important information's in the R1 corrected version. This manuscript is methodologically relevant and add some important information's in clinical virology practice.

The authors have taken into account all of the reviewer's remarks and queries. They have subsequently improved the proposed manuscript and they added important information's in the R1 corrected version. This manuscript is methodologically relevant and add some important information's in clinical virology practice.

The authors have taken into account all of the reviewer's remarks and queries. They have subsequently improved the proposed manuscript and they added important information's in the R1 corrected version. This manuscript is methodologically relevant and add some important information's in clinical virology practice.